# Do Greener Urban Streets Provide Better Emotional Experiences? An Experimental Study on Chinese Tourists

**DOI:** 10.3390/ijerph192416918

**Published:** 2022-12-16

**Authors:** Yanyan Zhang, Meng Wang, Junyi Li, Jianxia Chang, Huan Lu

**Affiliations:** 1School of Geography and Tourism, Shaanxi Normal University, Xi’an 710119, China; 2Shaanxi Key Laboratory of Tourism Informatics, Xi’an 710119, China

**Keywords:** tourists’ momentary emotional experience, unusual environment, urban green space (UGS), urban street, photo simulation, green view index (GVI) intervention

## Abstract

Compared to the usual environment, the potential momentary emotional benefits of exposure to street-level urban green spaces (UGS) in the unusual environment have not received much academic attention. This study applies an online randomized control trial (RCT) with 299 potential tourists who have never visited Xi’an and proposes a regression model with mixed effects to scrutinize the momentary emotional effects of three scales (i.e., small, medium and large) and street types (i.e., traffic lanes, commercial pedestrian streets and culture and leisure walking streets). The results identify the possibility of causality between street-level UGS and tourists’ momentary emotional experiences and indicate that tourists have better momentary emotional experiences when urban streets are intervened with large-scale green vegetation. The positive magnitude of the effect varies in all three types of streets and scales of intervention, while the walking streets with typical cultural attractions, have a larger impact relative to those with daily commute elements. These research results can provide guidance for UGS planning and the green design of walking streets in tourism.

## 1. Introduction

Researchers have been drawn to various issues that highlight the mental health effects related to unprecedented urbanization within the last decade [1]. Urban green space (UGS) is crucial in protecting mental health against many of the harmful impacts caused by rapid urbanization [2,3,4]. A considerable number of studies have listed the positive mental benefits of UGS from the perspective of environmental psychology [4,5,6]. Evidence from those studies indicates that a higher level of green space buffers depression, anxiety and stress [7,8,9] and promotes life satisfaction and happiness [10,11,12]. Especially amidst the doom and gloom of the COVID-19 pandemic, the fear of coronavirus has paralyzed life. The crisis may have caused mental affliction and grief due to the local and regional COVID-19 lockdowns [13]. It is vitally important to research the effects of UGS on citizens’ mental health and well-being.

UGS is not only comprised of recreational space, such as urban forests and public parks [8]. Rather, it is an important landscape element in urban environments—for instance, lawns, trees, shrubs and other forms of greenery [8,14,15]. In previous studies, urban spaces have been operationalized as lacking vegetation, and most focus on the positive relationship between large-scale green space (e.g., squares, urban parks) and mental benefits [8,16]. In fact, within urban streets, UGS is typically in the form of lawns or trees interlarded within roads and buildings [17]. Some studies show that even a single street tree may help to make an environment more pleasant [18]. Moreover, with the rapid urbanization of China, large-scale natural green spaces will undoubtedly shrink over time [19]. Streets are one of the most prevalent modes of landscape perception in urban cities, as they account for 25~35% of all developed urban land [19]. Urban streetscapes have become the most common type of daily landscape [19,20]. The green space of urban streets has more influence on mental well-being than similar spaces that are large in scale.

The methods adopted to measure green space play an essential role in studies of mental health–environment relations because the approach to calculating urban greenery largely influences the outcome of the research. Prior studies commonly adopted remote sensing to measure urban green space [15]. However, this approach omits detailed information on street greenery, especially smaller elements, including lawns and shrubs [21,22], which are important in shaping people’s eye-level perceptions of and experiences with UGS [21,22]. Some studies have found a positive relationship between mental health and green space by evaluating street view images [23]. The green view index (GVI), a standardized physical measure adopted to assess street-level visual greenery from pedestrians’ perspective, captures what people typically and actually see on the ground using photographs [15,24,25,26].

The measure of mental benefits is of vital importance in evaluating the value of UGS. Prior studies commonly focused on urban residents’ long-term well-being offered by the green space in the surrounding environment in which they live. For instance, White et al. [27] used a typical longitudinal sample of UK residents with individual and regional covariates controlled and found a positive relationship between higher levels of green space in living environments and lower levels of mental distress and higher life satisfaction. Indeed, considering people’s mobility in their daily lives [28,29], the findings of these studies fail to capture momentary experiences over time and place [30]. As a typical dynamic and mobile behavior, tourism is a component of daily urban life [31,32] that interrupts the ordinary human experience [31,33]. According to the concept of the unusual environment [34,35], there are significant differences between tourists’ short-term emotional experiences in the unusual environment and residents’ long-term well-being benefits of the usual environment [30,36,37]. It is also common to view life satisfaction from a person’s cognitive evaluation of life over a long period, while emotional evaluation focuses on the short-term [38,39]. Therefore, emotion, as a short-term [40] and intense reaction to stimulus in an individual’s environment [41], is useful as a psychographic variable [42] and can, therefore, be used to evaluate the beneficial effects of UGS, which tourists experience in the moment [43,44].

However, the prior studies have left several unanswered questions regarding tourists’ on-site emotional response to urban street-level green spaces in urban destinations while being away from their daily lives. Particularly, the most recent literature on tourism has pointed out that tourists’ increased search for green or outdoor spaces is related to the search for greater safety in the wake of the pandemic [45,46,47,48,49]. This is a vital question considering that the result has the potential to provide insight into the mechanisms behind tourists’ emotional experiences and can thus serve as a reference in planning higher-quality destinations in the shadow of the COVID-19 pandemic. To overcome these limitations and expand the research boundaries of UGS in an unusual environment setting, this study adopts photo simulation combined with pleasure–arousal (PA) dimensional measures to assess the effects of street-level GVI intervention on tourists’ momentary emotional experiences. This study hypothesizes that the emotional ratings of street view images will be positively affected by all three street types and scales. Furthermore, this study assumes that the magnitude of the effect will vary across the three street types and increase along the continuum of the GVI intervention scale.

### 1.1. Emotion and Green Spaces

Emotion is connected with how bodies inhabit and move through place [50], which is a result of a specific event or stimulus [51,52] and has become a core idea in tourism research [53]. It is considered critical to study tourists’ emotions as people make purchases according to their feelings [42]. The central role of emotion in the tourism experience has been empirically studied [42,43]. Some studies focused on spatiotemporal behavioral patterns in tourists’ emotional experiences and recognized several spatiotemporal behavioral patterns based on multisource data, from district-level to urban- and tourist attraction-level [54,55,56,57,58,59]. Other studies have investigated the influences of individual tourists’ emotions and explored their mechanisms by identifying two stimuli that elicit emotional responses in destination experiences, including social interaction factors [60] and the physical environment [61]. Several social interaction mechanisms of on-site emotion in the tourist experience have been proposed. In particular, evidence from multiple studies suggests that interplay with people is one trigger of emotions in tourism [43,62,63,64]. Other variables in the physical environment, such as weather [65], air quality [66,67] and urban forests [68], are related to direct emotions in the tourism experience. Emotional experiences are among the most important benefits achieved by exposing tourists to the natural environment [69].

In recent years, researchers have found that the urban built environment is an important source of stimuli that can evoke different emotional responses [70,71]. As visual stimuli provide the most information about the environment around us [72], visual exposure to urban space is one of the most important factors that affect people’s emotional experiences [71,73]. Stimulus-organism-response (SOR) offers a theoretical explanation for the emotional responses to urban space [61,70,74]. In line with the SOR theoretical framework, there are some experimental studies that have suggested that momentary exposure to green space may have a causal influence on people’s feelings [75]. For instance, evidence from an experimental study indicates that the affective outcomes of regions with greenery are significantly better than those without [76]. The findings from a street-scale green intervention study by Navarrete-Hernandez and Laffan [16] indicate that incorporating natural elements into individuals’ local environments can improve their happiness while reducing stress. This evidence suggests that the potential emotional benefits of higher levels of green space should be taken into consideration.

In order to accurately measure the potential emotional effects of controlled GVI intervention, this study manipulates the GVI using photo simulation to evaluate how tourists’ momentary emotional effects are triggered by street-level GVI intervention.

### 1.2. Photo Simulation

Vision is the principal sense used when walking on urban streets [19,53]. Emotional states have mostly been elicited using pictures as lead stimuli [77]. Studies on emotional perceptions often use photos of environments as visual stimuli [78]. This experimental method is able to ensure the internal validity of the presentation of the stimulus and identifies concrete causal inferences [79]. Photo simulations (PS), an approach to manipulating the physical environment captured by photographs, recreates changes to the visual stimuli used in the experiment. It has been broadly applied to assess the effects of green intervention on individual perceptions [16,80,81]. Several works have found that respondents rate photographs similarly to actual places, thus making photo simulation a cost-effective and preferred option to accurately measure people’s perceptions and attitudes [82,83]. This demonstrates that PS can be used to understand tourists’ emotional responses to UGS and, in particular, the street-level emotional experience. This study uses PS to design the urban street-level GVI intervention that stimulates tourists’ emotions and explores the relationship between urban street-level green space and tourists’ emotions associated with specific sites to provide suggestions for planning higher-quality destinations.

## 2. Methods

### 2.1. Experimental Design

This study is conducted through an online randomized control trial (RCT). Specifically, it is a 3 (street categories: traffic vs. commerce vs. culture) × 3 (intervention scale: small vs. medium vs. large) × 2 (treatment status: control vs. treatment) online experiment. Questionnaire Star (www.wjx.cn), a professional questionnaire survey website, is adopted in this survey to collect data on tourists’ emotional ratings of a set of digitally recreated photographs with different scales. Images are assigned to each tourist based on a double randomization process, which ensures that the covariates are balanced between participants and treatments. The order of images to be presented is randomized to control the potential fatigue and spillover effects. As the online RCT has an experimental group setting, each participant was randomly assigned either a control or treatment image to minimize research bias.

This experiment aims to examine whether street-level GVI intervention improves tourists’ momentary emotional experiences. Based on the internal reliabilities of tourists’ emotional scores of experimental images, a paired samples *t*-test and regression model are adopted to explore the effect of the intervention on tourists’ reported emotions. The paired samples *t*-test is applied to check the mean values between the control and treatment groups and test the significance of the difference in participants’ reported emotions between the two groups. A regression model is estimated to capture the average treatment effect of GVI intervention on participants’ reported emotions associated with urban streetscapes and explore the possibility of causality between the GVI intervention and tourists’ perceived momentary emotions.

### 2.2. Stimuli Construction

Xi’an is one of the well-known urban tourism cities in China and has become a popular destination for travelers. The urban center district, which is encircled by the city walls of the Ming Dynasty, is an area that is frequently visited [84,85]. Tourists visit the famous tourist attractions in the area, such as the Xi’an Bell Tower and Xi’an city wall, and have a stable emotional experience [55]. Accordingly, photos of three types of streets were chosen as visual stimuli for the experiment:

The photos were taken from 9:00 to 14:00 when the atmospheric lighting conditions are regulated and on clear or less cloudy days. The digital camera was placed approximately 1.50 m over the ground at horizontal eye level. The trained researcher stood on a sidewalk of a street to mimic what tourists will see and took photographs that captured the view parallel to the street. The photos were captured by a digital camera with a 4:3 aspect ratio and a focal length of 35 mm [19] with the same dimension (i.e., 3000 × 2000 pixels). Nine photos were selected covering the most attractive streetscapes of the three types of streets to ensure that respondents are immersed in the experiment.

In order to obtain a series of comparable photos, this study digitally manipulates a control version into a treatment version using Adobe Photoshop. All features were maintained, such as people, cars, and weather, among others, except for the intervention element being tested. The treatment version of the images was grouped into three scales of intervention: small, medium, and large. According to previous scholars [86,87,88], there are 4 or 5 levels of GVI of Chinese urban streetscapes, and the difference between the levels are 10%, 20% and 5%. At the street level, this study chooses the 5% difference to calculate GVI effectively. Therefore, the three scales of GVI interventions were 5~9.9%, 10~14.9% and 15~20%. Nine sets of control and treatment images were created in this study. The stimuli settings are displayed in Figure 1.

### 2.3. Measurement of Emotion

To ensure the accuracy of the results, it is of considerable importance to select an appropriate measurement approach to measure individuals’ emotional responses to the images with and without interventions. The pleasure-arousal-dominance (PAD) framework is employed to measure tourists’ emotional responses to 18 images (nine sets of control and treatment images). PAD is the most common theory within the dimensional framework [74]. It was initially applied to theories of environmental psychology, the core concept of which is that the physical environment impacts people’s emotions [74]. The dimensional model has been widely used in emotion recognition experiments due to its ability to locate discrete emotions without a specific category definition [89]. However, because dominance identifies very little variance in emotions, it was deleted from the PAD model in Russell’s later research [90,91]. The PA dimension model is widely used in later studies and has been demonstrated to be a reliable approach to measuring tourists’ emotional states in different types of cultures and environments [92,93]. Pleasure reflects the degree to which the individual feels good about the surrounding environment, whereas arousal denotes the degree to which the person feels emotionally activated or stimulated [94]. In addition, safety and fearfulness [95,96] are considered independent factors in rating emotional responses to places and environments. As suggested by Russell and Pratt [91] and Hanyu [95,96], 8 items were selected to rate tourists’ emotional responses to urban street-level images associated with specific sites that represent the pleasure and arousal dimensions. Pleasure is measured using a 5-point semantic differential scale with the following four items: repulsive–nice, tense–restful, uncomfortable–comfortable and unpleasant–pleasant. Likewise, four items are used to measure arousal: monotonous–exciting, inactive–active, boring–interesting and fearful–safe.

### 2.4. Data Collection and Analysis

A pilot study was conducted both online and offline with 155 tourists from 15 January to 20 January 2020, covering potential tourists and those who had visited Xi’an. The internal reliability of emotional ratings was tested using IBM^®^ SPSS^®^ Statistics 25.0 (Armonk, NY, USA). However, the COVID-19 pandemic started in China after the initial set of questionnaire items improved. Given the enforced imposed mobility restrictions on a local, regional, and national scale, an online randomized control trial (RCT) was conducted to substitute the on-site experiment in this study. Questionnaire Star (www.wjx.cn), an online professional survey website, was applied to design the questionnaire. The visual stimuli were randomly assigned to participants through the situational random display mode on the Questionnaire Star. The situational random mode can import a variety of situational contexts, either text descriptions or photos and videos. The situations to be presented are randomized and assigned to participants by the online system [97]. An access link or a QR code for the experimental questionnaire will be generated.

A snowball sampling approach was deployed for data collection from February to March 2020. The online participants were invited through social media WeChat—a widely used social media platform in China. The online survey method has been successfully used in previous studies of mental well-being [19,98] and produced reasonable results in visual landscape assessment [99]. Prior tourism research has adopted WeChat to approach Chinese respondents, given its popularity in Chinese communities [13,100,101]. Moreover, it was ideal to use the online survey during the pandemic as the participants were protected from physical interactions. Firstly, the QR code of the online questionnaire was sent to a large group of WeChat users. A filter question was raised to ensure that the participants were potential tourists who had never visited Xi’an. Upon agreement, the participants were asked to imagine that they were walking along the street exhibited in the image and to rate the emotions of pleasure and arousal they associate with the site (on a scale of 1–5). Each participant was restricted to completing the survey only once and without material incentives. To avoid inducing fatigue, the order of the nine sets of control and treatment images was randomly assigned, and one image from each picture set was displayed for 10 s. The tourist could freely change his or her scores for any image before submitting the survey.

A total of 357 responses were received, and all respondents were Chinese. Fifty-eight respondents were removed from the dataset due to incompleteness or outlier responses. The drop rate is 16.25%. Of the 299 available respondents, 53.18% were male, and 46.82% were female. A total of 58.86% were between 18 and 25, and 18.39% were between 31 and 40 years of age. A total of 73.2% received up to a bachelor’s, master’s or Ph.D. degree, and the majority (73.24%) were not current students. Despite demographic distributions of gender and student status resonating closely with those reported in the online statistical reports [102,103], compared to the tourists who visited Xi’an in 2019 [104], generally speaking, there were more young participants than elderly, and there were more high-level educational participants than ones with low educational level (see Table 1). The reason may be that the research team who conducted the online surveys were young graduate students and staff in the university. They were easily connected to young people with high educational levels.

#### 2.4.1. Reliability

The mean value of all participants’ assessments for each item listed in the emotion scale is calculated, and the average score of eight items (i.e., 4 items representing the pleasure dimension and 4 items representing the arousal dimension) is used to measure the emotional responses to an urban street-level image. Indicator loadings are used to evaluate the reliability indicator. All items show significant standardized loadings over 0.68 (*p* < 0.001). In addition, the high Cronbach’s alphas indicate very good internal reliabilities for the overall emotion measurements and two dimensions (i.e., pleasure and arousal) based on the criterion developed by Landis and Koch [105]. The results are displayed in Table 2.

#### 2.4.2. Empirical Model

This study uses a regression model to quantitatively assess the average treatment impact by comparing the changes between the treatment and control groups. Each tourist rates 18 different images, which are explained by the random effects at the individual level. These reported ratings are not independent of each other. In addition, the fixed effects are included to control the average emotional ratings of each image. The model is as follows:*Emotion_ij_ = β_1_Treatment_i_ + β_2_Image_i_ + U_j_ + E_ij_*(1)
where the dependent variable, *Emotion_ij_*, indicates the tourists’ emotional scores given on a scale of 1–5, *j* indicates a single tourist, and *i* indicates the *i*th image to be presented to a tourist. *Treatment_i_*, our major independent variable of interest, is a dummy variable, where 1 (treatment) represents intervention being included in the *i*th image; otherwise, it is 0. The estimated coefficient of *Treatment* is reflected by *β*_1_ (the average treatment effect), which indicates the average change that the tourists report in terms of emotional rating for the images with intervention compared to those without. *Image_i_* represents the fixed effect of the *i*th image, reflected by *β*_2_. *U_j_* reflects the *j*th tourist’s random intercept. *E_ij_* represents the error term.

#### 2.4.3. Robustness Tests

To test the robustness of the results, control variables (i.e., sociodemographic variables), such as gender, age, employment status, student status and area of study, are included in the regression model. *X_ij_* represents the expectation that these variables will generate more reliable and robust results, reflected by *β*_3_. The adjusted model is expressed as follows:*Emotion_ij_ = β_1_Treatment_i_ + β_2_Image_i_ + β_3_X_ij_ + U_j_ + E_ij_.*(2)

SPSS 25.0 software was used to run Equations (1) and (2) in all versions. The results of these regressions are presented in the section below.

## 3. Results

### 3.1. The Impact of Overall Intervention on Tourists’ Momentary Emotional Experiences

In this section, an analysis of the full pooled sample is conducted to investigate whether the GVI intervention impacts tourists’ momentary emotional responses. Figure 2 shows the results of the paired samples *t*-test between the treatment and control groups. As illustrated in the graph, the differences in the means of tourists’ reported momentary emotional experiences are statistically significant, thus suggesting that tourists’ pleasure and arousal would be stronger on urban streets with higher GVI values. The average treatment effect of GVI intervention on tourists’ emotions is of prime importance. Figure 3 shows the estimated coefficient of *Treatment*. As observed in columns 1 and 3 of Table A1, higher levels of GVI significantly improve tourists’ reported pleasure and arousal. As shown in columns 2 and 4 of Table A1, incorporating socioeconomic controls does not influence the results, which suggests that GVI intervention has a positive impact on tourists’ momentary emotional experiences associated with urban streets.

### 3.2. The Effect of Three-Scale Intervention on Tourists’ Momentary Emotional Experiences

To examine whether larger-scale GVI intervention has a more powerful influence on the effect relative to smaller ones, an exploratory analysis was conducted to investigate the possibility of causality between the scale of GVI intervention and the average treatment effect of tourists’ momentary emotions. The mean difference of the paired samples *t*-test is statistically significant (see Figure A1 for details).

Figure 4 depicts the estimated coefficient of the three intervention scales on tourists’ pleasure and arousal. In terms of pleasure, the results suggest that the size of the average treatment effect increases with the scale of intervention. Columns 1, 3 and 5 of Table A2 show that the most powerful impact on pleasure is generated from large-scale intervention, followed by medium-scale intervention, and small-scale intervention has the smallest effect. While the arousal dimension indicates that the large- and medium-scale interventions have the same effect on arousal, the small scale has the smallest and displays a similar causality to pleasure (see Table A3 for details). These estimates are robust (see Table A2 and Table A3 for details).

### 3.3. The Impact of Three-Street Intervention on Tourists’ Momentary Emotional Experiences

In this section, this study will be extended to each individual street type to determine the momentary emotional effect of GVI intervention. In accordance with the research design in the above sections, both mean and coefficient estimations are provided. The mean difference of the paired samples *t*-test is statistically significant (see Figure A2 for details).

Figure 5 depicts the estimated coefficient for the three types of interventions on tourists’ pleasure and arousal, respectively. As seen in the graphs, the average treatment effect of GVI intervention by type has a significantly positive impact on tourists’ pleasure and arousal (see Table A4 and Table A5 for details). According to the magnitude of the coefficient, the positive impact of traffic lanes is the smallest. The greatest impact on pleasure is caused by the GVI intervention on walking streets, and the largest effect on reported arousal is caused by commercial pedestrian streets.

#### 3.3.1. Traffic Lanes

As Figure 6a shows, traffic lanes with a higher GVI have a greater impact on pleasure and arousal. As shown in columns 1 and 3 of Table A6, traffic lanes improve the reported levels of pleasure and arousal. These estimates presented in columns 2 and 4 of Table A6 remain robust after the inclusion of controls.

#### 3.3.2. Commercial Pedestrian Streets

Figure 6b reflects the average treatment effect of GVI intervention on tourists’ reported momentary emotions. According to the estimates displayed in column 1 of Table A7, the GVI intervention of commercial pedestrian streets significantly improves the reported levels of pleasure and arousal. As shown in columns 2 and 4 of Table A7, estimates of commercial pedestrian streets remain robust after including socioeconomic controls.

#### 3.3.3. Walking Streets

A strong positive impact was observed in culture and leisure walking streets, as shown in Figure 6c. As shown in columns 1 and 3 of Table A8, culture and leisure walking streets improve reported levels of pleasure and arousal. These results remain robust after the inclusion of individual socioeconomic controls (see Table A8 for details).

## 4. Discussion

The current study advances the prior studies by designing an unusual environment and expanding the research boundaries of UGS to a tourism setting. Previous studies have typically focused on the positive benefits of UGS on residents who live in the usual environment [27,106,107,108]. However, these studies failed to consider the momentary experiences stimulated by the unusual environment beyond tourists’ hometowns, as they are also the beneficiaries of green space in urban tourism destinations. The results indicate that an increase in urban green space can improve tourists’ momentary emotional experiences. This positive relationship is consistent with evidence from prior studies on residents’ long-term mental well-being [10,27]. Moreover, the current study responds to calls for a more rigorous study design and multidimensional measures of the relationship of UGS to mental well-being outcomes [3,4,109]. Primarily, it demonstrates that tourism destinations need greener, more balanced and more sustainable tourism in a COVID-19 and post-COVID-19 world [47], including mental/emotional sustainability [13].

Methodologically, this study adopts GVI to intervene in the green space ratio and strictly controls the visual greenery to manipulate experimental stimuli. Only by doing so can it accurately estimate the effect of intervention from pedestrians’ perspective. It is, therefore, more rigorous than Navarrete-Hernandez and Laffan [16] in terms of its construction of visual stimuli. The results demonstrate that the magnitude of the improvements in momentary emotional experience is effectively identified. Importantly, an increasing effect along the scale continuum of GVI intervention is confirmed. Larger GVI interventions have a more powerful impact on the average treatment effect, which indicates that street-level green space in unusual environments can be considered a trigger of tourists’ emotional experiences. As such, a possible empirical causality is proved between tourists’ emotional experience and street-level UGS, which introduces psychological environment theories to the study of the mechanisms behind tourists’ emotional experiences.

In terms of the operationalization of UGS’s form, this study finds that green space at the street level also contributes to its potential for generating emotional changes. Small green vegetation, such as trees, shrubs, and lawns, are included in the street images to intervene in green space at three different scales. Despite several of the nuances observed among intervention scales, the emotional effects are uniformly improved. These findings are in line with the prior studies [18] and provide empirical evidence of the links between momentary emotional experiences and urban street-level green space. It is also a response to the evidence that the nuanced relationship between UGS and emotional well-being may depend on the specific context of the greenery [17] and suggests that UGS interspersed among streets is of vital importance in the improvement of tourists’ emotional experience in tourism destinations.

GVI intervention is manipulated in three different streets, the impact of which differs according to the momentary emotional experience. All scales of GVI intervention on the three types of streets have a positive impact. While the greatest impact on reported pleasure is observed in culture and leisure walking streets, the largest effect on reported arousal is demonstrated in commercial pedestrian streets, and the smallest impact on pleasure and arousal is all observed in traffic lanes. These findings provide insight into the special effect of novelty in street environments. Tourists want novelty, being away from their routine or usual living environment [110] and staying away from mentally fatiguing conditions which are normal in their daily lives [17]. As such, culture and leisure walking streets, which have typical cultural attractions, have the largest impact. In contrast, streets with various daily commute elements, such as motor vehicles, show the smallest impact on the two dimensions of emotion. It appears that novelty (especially culture and historical elements) plays an important role in the relationship between UGS and tourists’ momentary emotional experiences and has an additional benefit in mental restoration in unusual environments [76].

Given the scale-related treatment effects in tourists’ emotional experiences measured by PA, it is perhaps surprising that we observed a different effect of interventions on SWB in three scales in a prior study. In this study, the positive impact of overall GVI intervention on emotion is consistent with prior studies [4,6,16]. However, the scale-related intervention effects are different from the result in Navarrete-Hernandez and Laffan [16]. Its greatest impact on perceptions of happiness is observed from medium-scale GVI intervention, while the magnitude of the average treatment effect on momentary emotional experience increases with the scale of the intervention in the current study, thus suggesting that the short-term emotions in the unusual environment react more intensely to stimulus. Additionally, this may be evidence that emotion measures are more appropriate for short-term visual simulation in the current study to evaluate the beneficial effects of UGS, which tourists are exposed to at the moment.

We now identify practical implications related to urban renewal, tourism planning and tourist satisfaction. First, the results indicate that some benefits of large-scale UGS can be provided by urban streets with small-scale greenery. Incorporating green vegetation into urban pedestrian streets is a successful strategy for improving tourists’ momentary emotional experiences. Hence, the green design of urban tourism streets should be considered in tourism planning. Second, policymakers and planners should consider the difference in impacts on momentary emotional experience among different types of streets and scales of intervention. Large-scale green vegetation should be incorporated into high-traffic streets to create better distinction and separation between the motorists’ environment and tourists. In addition, it is important to maintain cultural and historic features while designing greenery in cultural and leisure walking streets. Third, this method can be used to examine different greening planning designs for the renewal of historical walking streets.

## 5. Limitations and Suggestions for Further Studies

Some limitations should be addressed in further studies. First, this empirical study was conducted in China among Chinese participants. In light of the research context, findings must be interpreted with caution since transferability is limited. Further studies are encouraged to corroborate the findings in other cultural settings as well as to account for differences in other similar tourist cities in China. Second, the sample of this study resonates closely with the current WeChat user base. Although the statistical results remained robust after incorporating socioeconomic controls, it failed to recruit diversified participants, and future research is recommended to investigate various demographic groups. Third, this research is based on an online simulation, and the complex multisensory facets of real-world interactions cannot be captured in virtual exposures to visual stimuli. Other senses, such as soundscapes, may play an important role in the relationship between UGS and tourists’ momentary emotional experiences [111,112]. Future research can include other senses in a real-world street environment and recruit on-site tourists to further examine the impact of street-level UGS on momentary emotional experiences. Fourth, the approach of using photo simulation while controlling for the other contents of the image measures the impact of street-level GVI intervention on tourists’ momentary emotional experiences and cannot account for the effect of what is replaced by vegetation. It is possible that specific elements of the visual stimuli trigger the emotional state. Additionally, personal place attachments might have affected tourists’ emotional experience in spite of tourists being randomly assigned to the experimental groups [113]. Further studies should take qualitative research into consideration to offer useful insights to assess which specific elements of the experimental simulations (e.g., the mediating effect) will influence the outcomes. Last, the findings of this study may be subject to response bias and thus may not reflect an unbiased emotional state due to its use of immediate self-reported data. Actually, emotion is accompanied by physiological responses which are not controlled by the individual’s conscious [90]. Psychophysiological measurement devices are able to keep track of subjects’ physiological reactions to stimuli, thereby offering momentary information on a participant’s emotional reactions [114]. Consistent with the dimensional emotion approach, further studies can apply physiological measurements to provide measures of pleasure [115] and arousal [116], such as skin conductance (SC) and facial electromyography (EMG). Therefore, mixed-methods designs that also include qualitative research are especially recommended.

## 6. Conclusions

An online RCT with 299 potential Chinese tourists is carried out by combining photo simulation and PA dimensional emotion measures. This paper aims to assess whether urban street-level green space improves tourists’ momentary emotional experiences. Three key findings are summarized as follows: (1) street-level urban green space positively influences tourists’ momentary emotional experiences in all street types and GVI intervention scales; (2) the magnitude of the effect on emotional experience increases along the scale continuum of GVI intervention; (3) The positive magnitude of the effect observed in the walking streets which have typical cultural attractions has larger impact relative to those with daily commute elements. The results highlight the potential benefits of street-level UGS for enhancing tourists’ momentary emotional experiences and provide insight into the special effect of novelty in street environments.

## Figures and Tables

**Figure 1 ijerph-19-16918-f001:**
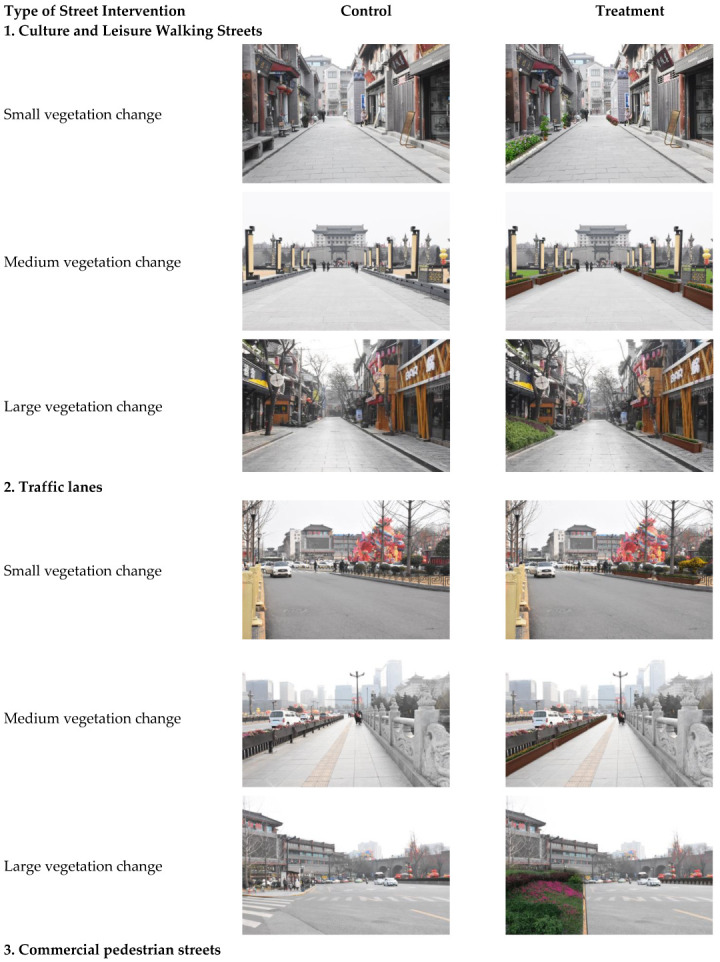
Visual stimuli.

**Figure 2 ijerph-19-16918-f002:**
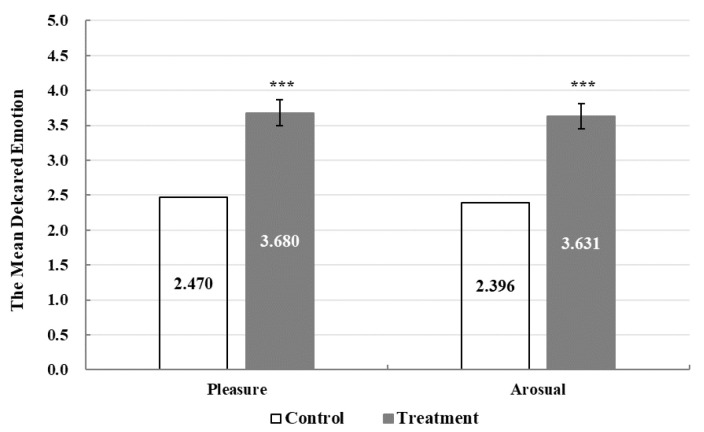
The impact of overall intervention on tourists’ momentary emotional experience. Note: Error bars represent 95% confidence interval. *** *p* < 0.001.

**Figure 3 ijerph-19-16918-f003:**
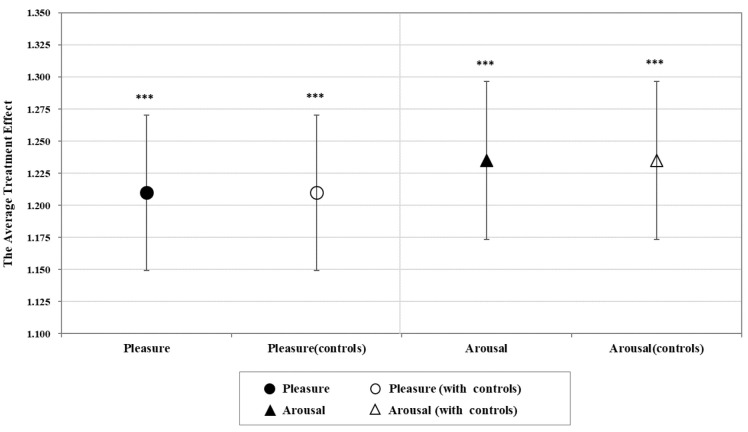
The effect of overall intervention on tourists’ momentary emotional experience. Note: Error bars represent 95% confidence interval. *** *p* < 0.001.

**Figure 4 ijerph-19-16918-f004:**
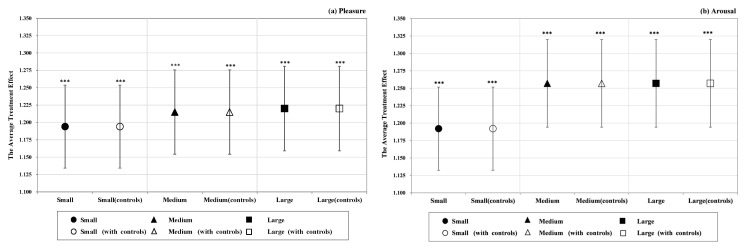
The effect of three scale intervention on tourists’ momentary emotional experience; (**a**) pleasure, (**b**) arousal. Note: Error bars represent 95% confidence interval. *** *p* < 0.001.

**Figure 5 ijerph-19-16918-f005:**
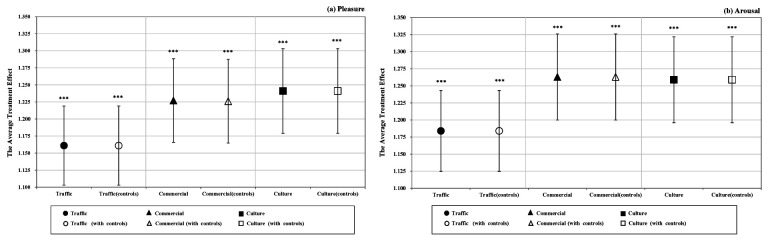
The effect of intervention on tourists’ pleasure and arousal; (**a**) pleasure, (**b**) arousal. Note: Error bars represent 95% confidence interval. *** *p* < 0.001.

**Figure 6 ijerph-19-16918-f006:**
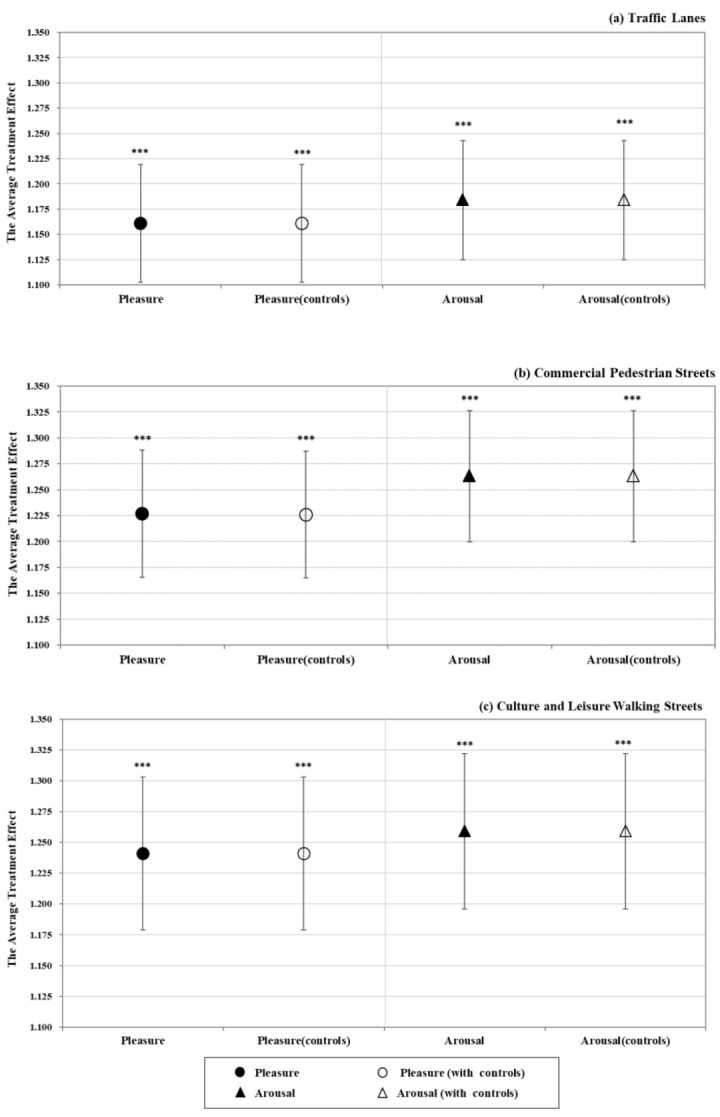
The effect of the three types of streets on tourists’ momentary emotional experience; (**a**) traffic lanes, (**b**) commercial pedestrian streets, (**c**) culture and leisure walking streets. Note: Error bars represent 95% confidence interval. *** *p* < 0.001.

**Table 1 ijerph-19-16918-t001:** Descriptive Statistics.

Variables	N	Mean	Std. Dev.	Min	Max	Frequency	Percentage (%)
Gender	5382	0.53	0.499	0	1		
Male						159	53.18
Female						140	46.82
Age	5382	28.68	8.302	18	51		
<18						0	0
18–25						176	58.86
26–30						27	9.03
31–40						55	18.39
41–50						40	13.38
51–60						1	0.33
>60						0	0
Student Status	5382	0.95	0.218	0	1		
Student						80	26.76
Not a student						219	73.24
Subject of study	5382	0.08	0.272	0	1		
Degree in Art, Design, etc.						24	8.03
No study in Art, Design, etc.						275	91.97
Educational level	5382	0.27	0.443	0	1		
Below a bachelor’s degree						15	5.02
Up to a bachelor’s degree						284	94.98

**Table 2 ijerph-19-16918-t002:** Factor loadings for each item.

Construct	Indicators	Loadings	Cronbach’s Alpha
Pleasure	Nice	0.740	0.947
	Restful	0.695	
	Comfortable	0.687	
	Pleasant	0.681	
Arousal	Exciting	0.822	0.950
	Active	0.783	
	Interesting	0.782	
	Safe	0.856	
The overall	-	-	0.972

## Data Availability

Data are contained within the article.

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
