# Peer review of "Do Greener Urban Streets Provide Better Emotional Experiences? An Experimental Study on Chinese Tourists"

_ijerph, 2022, doi:10.3390/ijerph192416918_

Round 1

Reviewer 1 Report

Firstly as a smaller issue, the title appears a bit generalised and long. It is just too many words in the current form for the article.

Some possibilities could be simply being more direct and specific in whom the territorial context is from the beginning and not trying to make it sound so universal.

Sometimes when you put too many specific concepts in the title it excludes readers who do not know they may find the piece interesting as it looks too specialised and presumes the knowledge of words they never have yet seemed defined before.

Remember, a title should not need so many definitions just to be understood.

Secondly, the most recent literature on tourism has pointed out that tourists' increased search for green or outdoor spaces is related to the search for greater safety in the wake of the pandemic. Particularly in China, where the COVID-19 pandemic has started, I believe this aspect needs to be over emphasized. On this aspect, some references that should be integrated are:

Casado-Aranda, L. A., Sánchez-Fernández, J., Bastidas-Manzano, A. B. (2021). Tourism research after the COVID-19 outbreak: Insights for More Sustainable, Local and Smart Cities. Sustainable Cities and Society, Vol. 73, 103126. https://doi.org/10.1016/j.scs.2021.103126

Ioannides, D., Gyimóthy, S. (2020). The COVID-19 crisis as an opportunity for escaping the unsustainable global tourism path. Tourism Geographies. Vol. 22(3), pp. 624-632. https://doi.org/10.1080/14616688.2020.1763445

Koh, E. (2020). The end of over-tourism? Opportunities in a post-COVID-19 world. International Journal of Tourism Cities. Vol. 4(6), pp. 1015-1023. https://doi.org/10.1108/IJTC-04-2020-0080 

Monaco S. (2020). Tourism, Safety and COVID-19. Security, Digitization and Tourist Behaviour. New York: Routledge.

The paper does not report the structure and the logic of the online tool applied. In addition, the reader is not provided with any useful information about the sample: the authors should clarify what recruitment procedures they followed, where they found respondents, what the drop rate was, what the biases and limitations were related to involving people who were recruited online (e.g., on the digital literacy of study participants). 

Finally, it would be good in considering internationalisation to ensure a line about transferability issues in the limitations discussion at the end of the piece. In considering how to write this, one asks is it likely the findings sets hold true for people in Europe, USA... ? Is this likely to occur say in Eastern countries but not other settings? Without too much detail, we need you to explain broadly how the study's context may or may not be similar to others in ways impacting transferability for some or all core findings as a general statement or two before the conclusion.

I think with these changes the piece would be an intresting addition to the journal. These changes are very important, but should be easily made.

Reviewer 2 Report

This paper successfully attempts to examine the effect of Urban Greening on tourists, building on previous research on Urban Green Spaces and their impact on well-being.  The methodological approach, as well as the interpretation and discussion of results are appropriate, relevant and presented clearly.  Perhaps, some transparency regarding the questionaire scales utilised, could render the paper even more useful for researchers intending to conduct further research on the topic.  Overall a well-researched, relevant and well-written study.  Congratulations to the authors.

Author Response

We deeply appreciate your positive evaluation of our work. Thank you for your congratulations.

Reviewer 3 Report

The paper shows an interesting overview of the topic of green urban space on the example of the Chinese tourist city Xi’an. Thanks to computer stimulation, the expected visualization of the target desired space is complemented with vegetation.

The paper is written in understandable language. The methodology, the results and the discussion are well presented in the paper. The statistics of the online surveys are also well presented in graphs and well described and do not give cause for complaint.

The research method presented in the manuscript can serve as a model for research in other regions of the world, including European cities, which are very often not green.
It would be interesting to compare the case study with another similar tourist city in China. However, I suggest that this subject be dealt with in further studies and articles, as the givrn case study is comprehensively presented in the manuscript submitted.
I suggest adding or specifying the final chapter of the conclusions.
I suggest also add information in the method chapter (2) about the computer software in which the statistics were made. This information now appears in chapter 3.

The manuscript meets the content and editorial requirements and is suitable for publication in the MDPI IJERPH.
